# MEMORY EFFICIENT DYNAMIC SPARSE TRAINING

## ABSTRACT

The excessive memory and energy consumption of modern Artificial Neural Networks (ANNs) is posing limitations on the machines that can run these models. Sparsification of ANNs is often motivated by time, memory and energy savings only during model inference, yielding no benefits during training. A growing body of work is now focusing on providing the benefits of model sparsification also during training. While these methods improve the energy efficiency during training, the algorithms yielding the most accurate models still have a peak memory usage on the same order as the dense model. We propose a Dynamic Sparse Training (DST) algorithm that reduces the peak memory usage during training while preserving the energy advantages of sparsely trained models. We evaluate our algorithm on CIFAR-10/100 using ResNet-56 and VGG-16 and compare it against a range of sparsification methods. The benefits of our method are twofold: first, it allows for a given model to be trained to an accuracy on par with the dense model while requiring significantly less memory and energy; second, the savings in memory and energy can be allocated towards training an even larger sparse model on the same machine, generally improving the accuracy of the model.

## 1 INTRODUCTION

Artificial Neural Networks (ANN) are currently the most prominent machine learning method because of their superiority in a broad range of applications, including computer vision (O'Mahony et al., 2019; Voulodimos et al., 2018; Guo et al., 2016a), natural language processing (Otter et al., 2020; Young et al., 2018), and reinforcement learning (Schrittwieser et al., 2020; Arulkumaran et al., 2017), among many others (Wang et al., 2020; Zhou et al., 2020; Liu et al., 2017). However, as ANNs keep increasing in size to further improve their representational power (Du et al., 2019; Novak et al., 2018), the memory and energy requirements to train and make inferences with these models becomes a limiting factor (Hwang, 2018; Ahmed & Wahed, 2020).

The scaling problem of ANNs is most prominent in the fully-connected layers, or the dense part of an ANN that includes multiplication with a weight matrix that scales quadratically with the number of units, making very wide ANNs infeasible. This problem is exacerbated as ANNs learn from high-dimensional inputs such as video and spatial data (Garcia-Garcia et al., 2018; Ma et al., 2019) and produce high-dimensional representations for many-classes classification or generative models for images and video (Mildenhall et al., 2021; Ramesh et al., 2021), all of which are gaining importance.

A large body of work has addressed the scaling problem (Reed, 1993; Gale et al., 2019; Blalock et al., 2020; Hoefler et al., 2021), many studies look into sparsity of the weight matrix as a solution based on the observation that the weight distribution of a dense model at the end of training often has a peak around zero, indicating that the majority of weights contribute little to the function being computed (Han et al., 2015). By utilizing sparse matrix representations and operations, the floating-point operations (FLOPS), and thus the energy usage, of a model can be reduced dramatically. Biological neural networks have also evolved to utilize sparsity, which is seen as an important property for learning efficiency (Pessoa, 2014; Bullmore & Sporns, 2009; Watts & Strogatz, 1998).

Early works in ANN sparsity removed connections, a process called *pruning*, of a trained dense model based on the magnitude of the weights (Janowsky, 1989; Ström, 1997), resulting in a more efficient model for inference. While later works improved upon this technique (Guo et al., 2016b; Dong et al., 2017; Yu et al., 2018), they all require at least the cost of training a dense model, yielding

no efficiency benefit during training. This limits the size of sparse models that can be trained on a given machine by the largest dense model it can train.

In light of this limitation, the Lottery Ticket Hypothesis (LTH) (Frankle & Carbin, 2018), surprisingly, hypothesized that there exists a subnetwork within a dense over-parameterized model that when trained with the same initial weights will result in a sparse model with comparable accuracy to that of the dense model. However, the proposed method for finding a *Winning Ticket* within a dense model is very compute intensive, as it requires training the dense model (typically multiple times) to obtain the subnetwork. Morover, later work weakened the hypothesis for larger ANNs (Frankle et al., 2019). Despite this, it was still an important catalyst for new methods that aim to find the Winning Ticket more efficiently.

Efficient methods for finding a Winning Ticket can be categorized as: pruning before training and Dynamic Sparse Training (DST). The before-training methods prune connections from a randomly initialized *dense* model (Lee et al., 2018; Wang et al., 2019; Tanaka et al., 2020). In contrast, the DST methods start with a randomly initialized *sparse* model and change the connections dynamically during training, maintaining the overall sparsity (Mocanu et al., 2018; Mostafa & Wang, 2019; Evci et al., 2020). In practice, DST methods generally achieve better accuracy than the pruning before training methods (Wang et al., 2019). In addition, the DST methods do not need to represent the dense model at any point, giving them a clear memory efficiency advantage, an important property for our motivation. The first DST method was Sparse Evolutionary Training (SET) (Mocanu et al., 2018). SET removes the connections with the lowest weight magnitude, a common pruning strategy (Mostafa & Wang, 2019; Dettmers & Zettlemoyer, 2019; Evci et al., 2020), and *grows* new connections uniformly at random. RigL (Evci et al., 2020) improved upon SET by growing the connections with the largest gradient magnitude instead. These connections are expected to get large weight magnitudes as a result of gradient descent optimization.

While these methods drastically reduce the FLOPS required to train sparse models, the pruning before training methods and RigL share an important limitation: *they have a peak memory usage during training on the same order as the dense model*. This is because the pruning before training methods use the dense randomly initialized model, and RigL requires the periodic computation of the gradient with respect to the loss for all possible connections. We present a DST algorithm that reduces the peak-memory usage to the order of a sparse model while maintaining the improvements in compute and achieving the same accuracy as RigL. We achieve this by efficiently sampling a subset of the inactive connections using a heuristic of the gradient magnitude. We then only evaluate the gradient on this subset of connections. Interestingly, the size of the subset can be on the same order as the number of active connections, therefore reducing the peak memory to the order of the sparse model. We evaluate the accuracy of our method on CIFAR-10/100 using ResNet-56 and VGG-16 models and compare it against a range of sparsification methods at sparsity levels of 90%, 95%, and 98%. The benefits of our method can be utilized in the following two ways:

1. It allows for a given model to be trained to an accuracy on par with the dense model while requiring significantly less memory and energy.

2. The savings in memory and energy can be allocated towards training an even larger sparse model on the same machine, generally improving the accuracy of the model.

## 2 RELATED WORK

A variety of methods have been proposed that aim to reduce the size of ANNs, such as dimensionality reduction of the model parameters (Jaderberg et al., 2014; Novikov et al., 2015), and weight quantization (Gupta et al., 2015; Mishra et al., 2018). However, we are interested in model sparsification methods because they reduce both the size and the FLOPS of a model. Following Wang et al. (2019), we categorize the sparsification methods as: pruning after training, pruning during training, pruning before training, and Dynamic Sparse Training.

**After training** The first pruning algorithms operated on dense trained models, pruning the connections with the smallest weight magnitude (Janowsky, 1989; Thimm & Fiesler, 1995; Ström, 1997; Han et al., 2015). This method was later generalized to first-order (Mozer & Smolensky, 1988; Karnin, 1990; Molchanov et al., 2019a;b) and second-order (LeCun et al., 1989; Hassibi & Stork,

1992) Taylor polynomials of the loss with respect to the weights. These methods can be interpreted as calculating an importance score for each connection based on how its removal will effect the loss (Guo et al., 2016b; Dong et al., 2017; Yu et al., 2018).

**During training**    Gradual pruning increases the sparsity of the model during training till the desired sparsity is reached (Zhu & Gupta, 2017; Liu et al., 2021a). Kingma et al. (2015) introduced variational dropout which adapts the dropout rate of each unit during training, Molchanov et al. (2017) showed that pruning the units with the highest dropout rate is an effective way to sparsify a model. Louizos et al. (2018) propose a method based on the reparameterization trick that allows to directly optimize the $L^0$-norm, which penalizes the number of non-zero weights. Alternatively, DeepHoyer is a differentiable regularizer with the same minima as the $L^0$-norm (Yang et al., 2019).

**Before training**    The Lottery Ticket Hypothesis (LTH) (Frankle & Carbin, 2018; Frankle et al., 2019) started a line of work that aims to find a sparse model by pruning a dense model before training (Liu et al., 2018). SNIP (Lee et al., 2018) uses the sensitivity of each connection to the loss as the importance score of a connection. GraSP (Wang et al., 2019) optimizes gradient flow to accelerate training, however, Lubana & Dick (2020) argue that preserving gradient flow instead is a better approach. Tanaka et al. (2020) highlight a problem in the aforementioned methods: they suffer from layer collapse in high sparsity regimes, that is, during the pruning phase all the connections of a layer can be removed, making the model untrainable. They propose SynFlow, which prevents layer collapse by calculating how each connection contributes to the flow of information using a path regularizer (Neyshabur et al., 2015b), similar to (Lee et al., 2019).

**Dynamic Sparse Training**    The methods in this last category, including ours, maintain the same sparsity throughout training but periodically rewire a fraction of the active connections. This involves periodically pruning a fraction of active connections, followed by growing the same number of inactive connections. SET (Mocanu et al., 2018) was the first method and used simple magnitude pruning and random growing of connections. DeepR (Bellec et al., 2018) assigns a fixed sign to each connection at initialization and prunes those connections whose sign would change during training. DSR (Mostafa & Wang, 2019) prunes connections with the lowest global weight magnitude and uses random growing, allowing the connections to be redistributed over the layers and used in layers where they contribute the most. SNFS (Dettmers & Zettlemoyer, 2019) improves upon the previous methods by using an informed growing criteria: it grows the connections with the largest gradient momentum magnitude. RigL (Evci et al., 2020) makes this more efficient by only periodically calculating the gradient with respect to all possible connections. We propose another efficiency improvement by only calculating the gradient periodically on a subset of the connections.

## 3    METHOD

While RigL (Evci et al., 2020) is currently the best DST method in terms of accuracy and efficiency, it has an important limitation: it evaluates the gradient magnitude with respect to the loss for *every possible connection* during the growing step. This means that it requires enough memory to store a gradient for every possible connection, which is the same as the memory requirement for storing the dense model. This is important because the memory requirements dictate the machine that can train a model. Our main contribution is a more efficient growing algorithm in terms of compute and memory. Specifically, we only calculate the gradients on a subset $\mathbb{S} \subset \mathbb{W}$ of all possible connections $\mathbb{W}$, illustrated in Figure 1. Periodically, the subset is randomly sampled during each growing step from a probability distribution based on a heuristic of the gradient magnitude. In this section we explain which heuristics were considered and how to efficiently sample from their distributions in the case of fully-connected layers. We provide a discussion in Appendix C on how to apply these heuristics to convolutional layers.

Perhaps the simplest heuristic is to sample the subset $\mathbb{S}$ from a uniform distribution over the inactive connections $\mathbb{W} \setminus \mathbb{A}$, where $\mathbb{A}$ denotes the set of active connections. A connection $e \in \mathbb{W}$ is fully specified by the input and output units it connects, i.e., $(a, b) = e$. Thus, to ensure an efficient sampling process, we can sample connections at random by independently sampling from distributions over the input and output units of a layer, which is equivalent to a joint probability distribution over all the connections in the layer. This ensures that the probability of sampling each connection

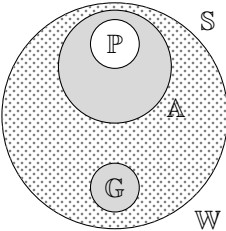 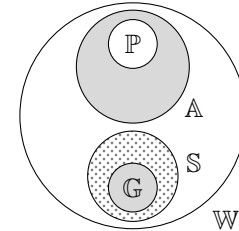

Figure 1: The illustrated set sizes of a pruning and growing round in RigL (left) and our method (right). Where $\mathbb{W}$ is the set of all possible connections, $\mathbb{A}$ the set of active connections, $\mathbb{P}$ the set of pruned connections, $\mathbb{G}$ the set of grown connections, and $\mathbb{S}$ the subset of inactive connections. The gray solid filled area becomes the set of active connections at the next time step.

is never explicitly represented, only the probability of sampling each unit is represented, which is more efficient. Formally, to create the subset we sample from the discrete uniform distributions over the input and output units of a layer $l$ such that the sampled connections are not active connections:

$$a_i \sim \mathcal{U}\left\{1, \ldots, n^{[l-1]}\right\}, \quad b_i \sim \mathcal{U}\left\{1, \ldots, n^{[l]}\right\}, \quad s.t. \ (a_i, b_i) \notin \mathbb{A} \tag{1}$$

where $n^{[l]}$ denotes the number of units in the $l$-th layer. Note that we abuse the notation of the various connection sets to be per layer or global interchangeably. Interestingly, sampling the subset uniformly at random in this way can be seen as the interpolation between SET and RigL, when $|\mathbb{S}|$ is equal to the size of the set of grown connections $|\mathbb{G}|$ it simplifies to SET and when $|\mathbb{S}|$ is equal to the number of possible connections $|\mathbb{W}|$ it simplifies to RigL.

In general, we sample the endpoints of the connections in the $l$-th layer from the discrete probability distributions $a_i \sim \mathrm{f}^{[l]}$ and $b_i \sim \mathrm{g}^{[l]}$ such that the sampled connections are not active connections. In this general setting, the distributions f and g don't need to be uniform, it might instead be useful to bias the subset towards connections that are more likely to have a large gradient magnitude and therefore serve as a better heuristic. To this end, we investigate two other distributions: the first uses the gradient of synaptic flow which measures how much an inactive connection would contribute to the overall connectivity of the model; and the second uses an upper bound of the gradient magnitude. In Section 4.2 we assess which of these heuristics is most appropriate for training sparse models.

## 3.1 SYNAPTIC FLOW (SYNFLOW)

First introduced by Neyshabur et al. (2015b), the path regularizer measures the contribution of each connection to the overall synaptic flow, for example, a given connection contributes more when it has strong afferent and efferent connections, even though its weight might not be particularly large. This regularizer was used to train unbalanced ANNs more effectively (Neyshabur et al., 2015a). In the context of model sparsification, Tanaka et al. (2020) used the path regularizer $R(\theta)$ in Equation 2. Their pruning before training algorithm, named SynFlow, prunes connections using a generalization of the weight magnitude $M(\theta)$ with the goal of preventing layer collapse. We use $\theta$ to denote all the parameters and use $\theta^{[l]}$ as the notation for the sparse weight matrix of the $l$-th layer.

$$R(\theta) = \mathbf{1}^\mathsf{T} \left(\prod_{l=1}^{L} \left|\theta^{[l]}\right|\right) \mathbf{1}, \quad M(\theta) = \frac{\partial R(\theta)}{\partial \theta} \odot \theta \tag{2}$$

Our motivation differs from that of Tanaka et al. (2020). We are interested in SynFlow because the connectivity of a unit can be used as a heuristic of the gradient magnitude. Intuitively, a connection with strong synaptic flow to its input unit and from its output unit is more likely to have a large gradient magnitude. This is due to the gradient information propagating over those same strong connections. From the perspective of a single layer, we sample an input unit proportionally to its afferent synaptic flow and an output unit proportionally to its efferent synaptic flow as follows:

$$\mathrm{f}^{[l]} := \frac{\mathbf{1}^\mathsf{T}\left(\prod_{k=1}^{l-1}\left|\theta^{[k]}\right|\right)}{\mathbf{1}^\mathsf{T}\left(\prod_{k=1}^{l-1}\left|\theta^{[k]}\right|\right)\mathbf{1}}, \quad \mathrm{g}^{[l]} := \frac{\left(\prod_{k=l+1}^{L}\left|\theta^{[k]}\right|\right)\mathbf{1}}{\mathbf{1}^\mathsf{T}\left(\prod_{k=l+1}^{L}\left|\theta^{[k]}\right|\right)\mathbf{1}} \tag{3}$$

The SynFlow heuristic only requires a single forward and backward pass to evaluate the afferent and efferent synaptic flow for all layers. This is achieved by creating a copy of the model whose weights take the absolute value of the original weights and passing it an all ones input vector. In addition, the nonlinearities of this model copy need to be linear in the positive domain.

## 3.2 GRADIENT MAGNITUDE UPPER BOUND (GRABO)

Since we are ultimately interested in finding the inactive connections with the largest gradient magnitude, sampling connections with a probability proportional to their gradient magnitude would be an ideal heuristic. The gradient magnitude for a single connection can be expressed as follows:

$$\left|\nabla\theta^{[l]}\right| = \left|\left(\boldsymbol{h}^{[l-1]}\right)^{\mathsf{T}}\boldsymbol{\delta}^{[l]}\right| \implies \left|\nabla\theta_{a,b}^{[l]}\right| = \left|\sum_{i=1}^{B}\boldsymbol{h}_{i,a}^{[l-1]}\boldsymbol{\delta}_{i,b}^{[l]}\right| \tag{4}$$

where $\boldsymbol{h}^{[l]}$ and $\boldsymbol{\delta}^{[l]}$ are the activation and gradient of the loss at the output units of the $l$-th layer, respectively. This simplifies to $\left|\boldsymbol{h}_{1,a}^{[l-1]}\boldsymbol{\delta}_{1,b}^{[l]}\right|$ when the batch size $B$ is 1. The simplified expression can then be used to sample connections efficiently according to:

$$\mathrm{f}^{[l]} := \frac{\left|\boldsymbol{h}_{1,\cdot}^{[l-1]}\right|}{\mathbf{1}^{\mathsf{T}}\left|\boldsymbol{h}_{1,\cdot}^{[l-1]}\right|}, \quad \mathrm{g}^{[l]} := \frac{\left|\boldsymbol{\delta}_{1,\cdot}^{[l]}\right|}{\mathbf{1}^{\mathsf{T}}\left|\boldsymbol{\delta}_{1,\cdot}^{[l]}\right|} \tag{5}$$

This induces a joint probability distribution that is proportional to the gradient magnitude only when the batch size is 1. In practice, however, training samples come in mini batches during gradient descent in order to reduce the variance of the gradient. This means that Equation 5 does not induce the correct distribution.

Instead, we sample connections proportionally to the following upper bound of the gradient magnitude which enables efficient sampling, even with mini batches:

$$\left|\nabla\theta^{[l]}\right| = \left|\left(\boldsymbol{h}^{[l-1]}\right)^{\mathsf{T}}\boldsymbol{\delta}^{[l]}\right| \leq \left(\sum_{b}^{B}\left|\boldsymbol{h}_{b,\cdot}^{[l-1]}\right|\right)^{\mathsf{T}}\left(\sum_{b}^{B}\left|\boldsymbol{\delta}_{b,\cdot}^{[l]}\right|\right) \tag{6}$$

The proof for Equation 6 involves the triangle inequality and is provided in Appendix D. Connections are then sampled from the following probability distributions:

$$\mathrm{f}^{[l]} := \frac{\sum_{i}^{B}\left|\boldsymbol{h}_{i,\cdot}^{[l-1]}\right|}{\sum_{i}^{B}\mathbf{1}^{\mathsf{T}}\left|\boldsymbol{h}_{i,\cdot}^{[l-1]}\right|}, \quad \mathrm{g}^{[l]} := \frac{\sum_{i}^{B}\left|\boldsymbol{\delta}_{i,\cdot}^{[l]}\right|}{\sum_{i}^{B}\mathbf{1}^{\mathsf{T}}\left|\boldsymbol{\delta}_{i,\cdot}^{[l]}\right|} \tag{7}$$

This particular upper bound has the property that Equation 7 simplifies to Equation 5 when $B = 1$. The implementation of this heuristic has the advantage that it does not require any modifications to the model, as is the case with the SynFlow heuristic. Yet it still only requires a single forward and backward pass to evaluate the distributions.

## 4 EXPERIMENTS

At this point, the foremost question is: *what is the minimum size of the subset that maintains the maximum accuracy?* We answer this question in Section 4.2 and then compare the accuracy of our method against other sparsification methods in Section 4.3. Lastly, in Section 4.4 we explore the relation between the width of a model and its accuracy while keeping the number of active connections constant across all model widths by increasing the sparsity of wider models.

### 4.1 EXPERIMENT SETUP

We evaluate our method and compare it to baselines on the CIFAR-10 and CIFAR-100 datasets (Krizhevsky et al., 2009). To establish how each method compares across model architectures we experiment with ResNet-56 (He et al., 2016) and use a 4 times downscaled version of VGG-16

(Simonyan & Zisserman, 2014) such that both models have roughly the same number of parameters. The bias and batch-norm parameters are kept dense since they only contribute marginally to the size of the model. We repeat each experiment 3 times across different seeds and report the mean and plot the 95th percentile. We adopt the implementation of the baselines based on their available code.

All experiments use the same optimization settings in order to isolate the differences in sparsification. Similar to Evci et al. (2020) and Lubana & Dick (2020), we use SGD with a momentum coefficient of 0.9, an $L^2$ regularization coefficient of 0.0001, and an initial learning rate of 0.1 which is dropped by a factor of 10 at epochs 80 and 120. We use a batch size of 128 and train for a maximum of 200 epochs. Training stops early when the loss does not improve for 50 epochs. We also apply standard data augmentation to the training data, including random flips and crops.

The LTH procedure described by Frankle & Carbin (2018), denoted as Lottery in the results, uses iterative pruning with iterations at 0%, 50%, 75%, 90%, 95% and 98% sparsity. The gradual pruning method by Zhu & Gupta (2017), denoted as Pruning in the results, reaches the target sparsity at the second learning rate drop and prunes connections every 1000 steps. The pruning before training methods use one iteration over all the training data to prune the dense model to the target sparsity. All the DST methods use the same update schedule: the connections are updated every 1000 steps, and similar to Evci et al. (2020), the fraction of active connections that is pruned is cosine annealed from 0.2 to 0.0 at the second learning rate drop. This is because Liu et al. (2021a) showed that DST methods struggle to recover from pruning when the learning rate is low.

It is important to note, given our motivation, that the DST methods initialize the sparse model in a way that does not require to represent or store the dense model at any time. In particular, each layer is initialized as a random bipartite graph using the Erdős–Rényi $G(n, M)$ random graph generation algorithm (Erdős & Rényi, 1959). The weights of the random sparse model are initialized using the procedure proposed by Evci et al. (2019) which adapts standard weight initialization to take into account the actual fan-in of each unit in order to properly scale the weight distribution for each unit.

Although the sparsity at initialization is assigned uniformly to all the layers, the pruning and growing procedures are applied globally. This means that the pruned connections are dynamically redistributed across layers throughout training, maintaining the overall sparsity. This approach generally outperforms local, layer-wise sparsity (Mostafa & Wang, 2019). To ensure stable training we found that it is important to sample the subset layer-wise, i.e., the subset size of a layer is proportional to the size of $\mathbb{W}$ of that layer. The grown connections are then taken globally again as the connections among all subsets that have the largest gradient magnitude.

## 4.2 SIZE OF THE SUBSET

First, we want to determine how the size of the subset affects the accuracy. In addition, we are interested in comparing the accuracy obtained by the three heuristics. The size of the subset is set proportionally to the number of pruned connections $|\mathbb{P}|$, with $|\mathbb{S}|$ being $\{1, 2, 4, 6, 8, 10\}$ times $|\mathbb{P}|$. Note that when $|\mathbb{S}| = |\mathbb{P}| = |\mathbb{G}|$, i.e., when the multiplier is one, our method is identical to SET in the case of the random heuristic. The results are shown in Figure 2 where the size of the subset is expressed as a fraction of $|\mathbb{W}|$ instead. The results include the baseline accuracy of SET and RigL. The extended results in Appendix E show the same trends and include 90% and 95% sparsity.

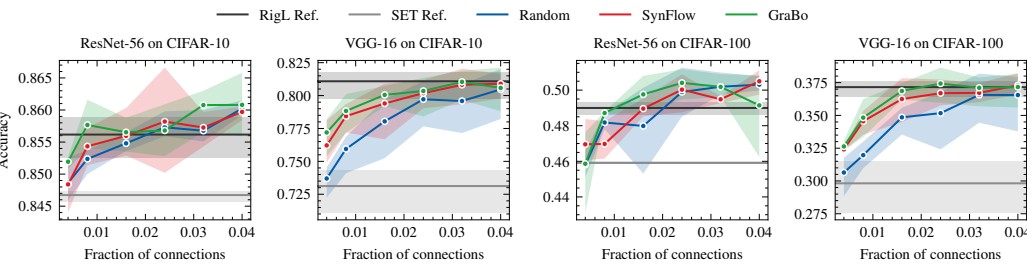

Figure 2: The accuracy of the heuristics while increasing the subset size $\mathbb{S}$ as a fraction of $|\mathbb{W}|$. Showing ResNet-56 and VGG-16 models trained on the CIFAR-10/100 datasets with 98% sparsity.

Remarkably, it is enough for the subset to contain only 3% of all the connections and still consistently achieves the accuracy of RigL. This is on the same order of connections as the number of active connections, thus reducing the peak memory usage to that on the order of the sparse model. The same trend appears among all the heuristics, but in general the GraBo heuristic reaches the maximum accuracy at smaller subset sizes. The SynFlow heuristic, in turn, outperforms the Random heuristic, indicating that our method indeed benefits from using informed heuristics. Interestingly, with ResNet-56 our method outperforms RigL when the subset size contains 4% of all the connections, however, this trend is not present for VGG-16 nor does it appear at lower sparsity levels where the accuracy of our method is on par with RigL.

## 4.3 COMPARISON WITH RELATED WORK

We also compare our method to a broad range of sparsification methods in Table 1. The Random method denotes training a static random sparse model, and SynFlow (Tanaka et al., 2020) is the pruning before training method, not to be confused with our method and the SynFlow heuristic. The other baselines are Lottery (Frankle & Carbin, 2018), Pruning (Zhu & Gupta, 2017), SNIP (Lee et al., 2018), GraSP (Wang et al., 2019), |GraSP| (Lubana & Dick, 2020), SET (Mocanu et al., 2018), and RigL (Evci et al., 2020). Our results in Table 1 use the GraBo heuristic with the subset size being 10 times the number of pruned connections.

Table 1: Top-1 accuracy of ResNet-56 and VGG-16 on CIFAR-10 and CIFAR-100

| Dataset | CIFAR-10 | | | | | | CIFAR-100 | | | | | |
|---|---|---|---|---|---|---|---|---|---|---|---|---|
| Model | ResNet-56 | | | VGG-16 | | | ResNet-56 | | | VGG-16 | | |
| Dense | 91.87 | | | 89.31 | | | 68.80 | | | 62.50 | | |
| Sparsity | 90% | 95% | 98% | 90% | 95% | 98% | 90% | 95% | 98% | 90% | 95% | 98% |
| Lottery | 89.18 | 87.90 | 84.40 | 87.82 | 86.12 | 82.42 | 61.58 | 57.44 | 48.82 | 53.67 | 48.10 | 38.36 |
| Pruning | 90.20 | 89.09 | 86.71 | 88.52 | 87.70 | 86.20 | 66.62 | 63.77 | 57.13 | 57.89 | 55.20 | 45.93 |
| Random | 88.85 | 86.88 | 73.98 | 83.11 | 77.59 | 58.95 | 60.68 | 51.64 | 31.43 | 46.83 | 37.39 | 21.29 |
| SNIP | 89.89 | 88.29 | 84.67 | 87.15 | 84.19 | 72.78 | 62.15 | 56.33 | 43.48 | 52.80 | 45.85 | 16.09 |
| GraSP | 87.48 | 86.58 | 81.88 | 85.97 | 84.58 | 80.24 | 58.13 | 52.78 | 37.83 | 51.56 | 46.45 | 35.19 |
| |GraSP| | 89.19 | 87.51 | 82.04 | 85.89 | 82.10 | 34.73 | 42.60 | 22.38 | 11.74 | 52.04 | 44.25 | 01.00 |
| SynFlow | 89.70 | 88.42 | 85.14 | 87.72 | 85.78 | 80.03 | 61.79 | 54.06 | 35.27 | 53.04 | 47.25 | 35.02 |
| SET | 89.89 | 88.29 | 84.67 | 86.10 | 83.34 | 73.12 | 63.90 | 58.04 | 45.91 | 52.77 | 44.82 | 29.82 |
| RigL | 90.55 | 89.29 | 85.62 | 88.33 | 86.24 | 81.09 | 64.91 | 59.88 | 49.00 | 54.82 | 49.01 | 37.16 |
| *Ours* | 90.56 | 89.29 | 86.08 | 87.80 | 86.14 | 80.58 | 64.72 | 60.05 | 49.14 | 54.76 | 48.68 | 37.17 |

Among the sparse training methods (from Random downwards in Table 1) our method and RigL outperform the other methods consistently over all datasets, sparsities, and model architectures. While at 90% sparsity all the methods achieve comparable accuracy, at the extreme sparsity rate of 98%, differences between methods become more evident. We see that gradually pruning connections during training (Pruning) can improve the accuracy quite significantly, especially in the extreme sparsity regimes. However, this comes at the cost of training the dense model, thus requiring significantly more memory and energy. We see this as motivation for research into sparse training methods which further increases their accuracy while preserving their efficiency advantage.

## 4.4 THE VALUE OF DIMENSIONS

In our final experiment we investigate how the width of the model affects the accuracy when the number of active parameters is kept constant. The width is used to multiply the number of filters in convolution layers and units in fully-connected layers. The results are presented in Table 2.

While basic machine learning theory tells us that the complexity of a function is directly related to the number of parameters, the results indicate that there are ways to use a certain parameter budget more effectively. Another classical example of this is the convolutional layer, its success stems from its parameter efficiency by consistently outperforming fully-connected layers with the same number of parameters in vision tasks. However, an important distinction with sparse methods is that there is

Table 2: Effect of network width on CIFAR-100

| Model | | ResNet-56 | | | | VGG-16 | | |
| Width | $|\mathbb{W}|$ | Sparsity | Top-1 acc. | | $|\mathbb{W}|$ | Sparsity | Top-1 acc. |
|-------|------|----------|------------|--|------|----------|------------|
| 1.0 | 0.9M | 0% | 68.80 | | 0.9M | 0% | 62.50 |
| 1.5 | 1.9M | 55% | 71.46 | | 2.1M | 55% | 65.86 |
| 2.0 | 3.4M | 75% | 72.07 | | 3.8M | 75% | 67.18 |
| 2.5 | 5.3M | 84% | 72.56 | | 5.9M | 84% | 67.86 |
| 3.0 | 7.6M | 89% | 72.21 | | 8.4M | 89% | 68.37 |
| 3.5 | 10.4M | 92% | 72.75 | | 11.5M | 92% | 68.32 |
| 4.0 | 13.6M | 94% | 72.70 | | 15.0M | 94% | 68.45 |

no inherent bias from human design, i.e., the active connections are found as a result of the training process and not hand crafted as is the case with convolutional layers.

Moreover, the results indicate that enabling training of larger models on the same machine because of the memory and compute savings of our method translates to increased accuracy (up to a certain sparsity level). These findings are similar to those from Zhu & Gupta (2017), however, our method enables a sparse model to be trained whose dense version would not fit on a given machine, enabling any given machine to train more accurate models.

## 5 DISCUSSION

We conjecture that the higher accuracy achieved by gradual pruning (Zhu & Gupta, 2017) is, at least partly, because it considers the entire connection search space, whereas our method and RigL use a greedy exploration algorithm, the gradient magnitude. This topic has been investigated by Liu et al. (2021a;b), who improve the performance of RigL by increasing the search space. We consider these extensions orthogonal to our work as they benefit from incorporating our method as well.

Mocanu et al. (2018) and Evci et al. (2020) differentiate between three sparsity distribution strategies: uniform, Erdős–Rényi, and Erdős–Rényi-Kernel. The later two are for fully-connected and convolutional layers, respectively, and assign a relative higher sparsity to layers with many possible connections. For simplicity, our experiments use a uniform sparsity distribution but we note that our method works with any of the improved sparsity distributions.

Like many other methods, we use magnitude pruning throughout training. With pruning before training, layer collapse has been observed as a result of magnitude pruning when a model is pruned to high levels of sparsity. Tanaka et al. (2020) investigated layer collapse and found that the training process is implicitly regularizing the magnitudes of the connections which minimizes the risk of layer collapse during DST. In addition, magnitude pruning is also very efficient, finding the top-k among a set of weights has a complexity of $O(|\mathbb{A}| \log |\mathbb{P}|)$.

Like Dettmers & Zettlemoyer (2019), our implementation is build using PyTorch and uses masked weights to simulate sparse neural networks. This is because support for sparse matrix operations in machine learning frameworks is currently incomplete. We therefore cannot provide any concrete measurements on memory savings. We leave it to follow up studies to implement our method in an efficient sparse manner to evaluate the practical memory savings of our method. Our work focuses on the theoretic and algorithmic aspects of a more memory efficient dynamic spare training method.

Lastly, we provide plots of the layer-wise sparsity of every method on each model, dataset, and at sparsity levels of 90%, 95%, and 98%. This information provides insight into how each method distributes the available connections over the layers of a model. Since this data is not essential to the conclusion of our paper we provide them as supplementary material in Appendix F.

## 6 CONCLUSION

We present a dynamic sparse training algorithm that reduces the peak memory during training while maintaining the same accuracy as the current best methods. This is achieved by only evaluating the

gradient for a subset of the connections instead of all possible connections. Our method achieves the same accuracy while evaluating the gradient for only 3% of the connections at 98% sparsity. The connections in the subset are sampled proportionally to a heuristic of the gradient magnitude. We compare three heuristics: uniform, the gradient of synaptic flow, and the upper bound of the gradient magnitude. We find that the upper bound of the gradient magnitude generally reaches the maximum accuracy at smaller subset sizes. In addition, the GraBo heuristic is also easier to implement as it doesn't require a model copy, like the SynFlow heuristic, therefore we conclude that the GraBo heuristic is the most appropriate for sparse training. Lastly, we show that the savings in memory and compute can be allocated towards training even wider models to achieve better accuracy. Our method reduces the peak memory usage to the order of a sparse gradient update, enabling a given model to be trained on a machine with less memory.

### REPRODUCIBILITY

To ensure reproducability we used standard model architectures and stated any modifications in Section 4.1. We used the CIFAR-10/100 datasets with their train and test split as provided by the Torchvision library. The optimizer settings are stated in Section 4.1. The data normalization and augmentation settings are specified in Appendices A and B, respectively. We provide a proof of the gradient magnitude upper bound used in Section 3.2 in Appendix D. The procedure for sampling the subset in convolutional layers is described in Appendix C. Additional implementation details are provided in Appendix G. Our aim is to open source the code for the experiments but as of the paper deadline this is pending legal approval.

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

## A DATA NORMALIZATION

We normalize all the training and test data to have a mean of 0 and a standard deviation of 1. The dataset statistics that we used are specified below, where $\mu$ is the mean and $\sigma$ the standard deviation. Each value corresponds to a color channel of the images.

### A.1 CIFAR-10

$$\mu = (0.4914, 0.4822, 0.4465), \quad \sigma = (0.2470, 0.2435, 0.2616)$$

### A.2 CIFAR-100

$$\mu = (0.5071, 0.4865, 0.4409), \quad \sigma = (0.2673, 0.2564, 0.2762)$$

## B DATA AUGMENTATION

We used standard data augmentation as part of our training data pipeline. The specific augmentation techniques are specified below.

### B.1 CIFAR-10

We use a random horizontal flip of the image with a probability of 0.5. We then pad the image with 4 black pixels on all sides and randomly crop a section of 32 by 32 pixels.

### B.2 CIFAR-100

We pad the image with 4 black pixels on all sides and randomly crop a section of 32 by 32 pixels. We then randomly flip the image horizontally with a probability of 0.5. Lastly, we randomly rotate the image between 15 degrees clockwise and counter clockwise.

## C CONVOLUTIONAL LAYER SUBSET SAMPLING

While in fully-connected layers a weight is specified by the input and output units that it connects, a weight in a 2D convolutional layer is specified by the input channel, output channel, and its $x$ and $y$ coordinate on the kernel. This means that it requires four discrete probability distributions to sample a weight of a 2D convolutional layer, instead of two for fully-connected layers.

With the uniform heuristic these four distributions are simply discrete uniform distributions between 1 and the number of input channels, output channels, kernel width, and kernel height. With the SynFlow heuristic, we sample an input channel proportional to the sum of afferent synaptic flow over the input channels and an output channel proportional to the sum of efferent synaptic flow over the output channels. With the GraBo heuristic, we sample an input channel proportional to the sum of activity over the input channels and an output channel proportional to the sum of the gradient over the output channels. The $x$ and $y$ coordinate on the kernel are more challenging to sample efficiently with the informed heuristics, therefore we decided to keep sampling those uniformly.

## D GRADIENT MAGNITUDE UPPER BOUND PROOF

**Lemma D.1.** *The gradient magnitude of the loss with respect to the parameters of a fully-connected layer $l$ has the following upper bound:*

$$\left| \nabla \theta^{[l]} \right| \leq \left( \sum_i^B \left| \boldsymbol{h}_{i,\cdot}^{[l-1]} \right| \right)^{\mathsf{T}} \left( \sum_i^B \left| \boldsymbol{\delta}_{i,\cdot}^{[l]} \right| \right)$$

where $\theta^{[l]} \in \mathbb{R}^{n^{[l-1]} \times n^{[l]}}$, $\boldsymbol{h}^{[l]} \in \mathbb{R}^{B \times n^{[l]}}$, and $\boldsymbol{\delta}^{[l]} \in \mathbb{R}^{B \times n^{[l]}}$ are the sparse weight matrix, activation and gradient of the loss at the output units of the l-th layer, respectively. The number of units in the l-th layer is denoted by $n^{[l]} \in \mathbb{N}^+$, and $B \in \mathbb{N}^+$ is the batch size.

*Proof.* The proof starts with the definition of the gradient of the fully-connected layer, which is calculated during back propagation as the activations of the previous layer multiplied with the back propagated gradient of the output units of the layer. We then write this definition for a single connection in the weight matrix and use the triangle inequality to specify the first upper bound. In Equation 10 we rewrite the upper bound of Equation 9 as all the cross products between the batch dimensions minus all the non-identical cross terms. Not that all the non-identical cross terms are positive because of the absolute, therefore we get another upper bound by removing the non-identical cross terms which we then write in matrix form in Equation 11.

$$\left| \nabla \theta^{[l]} \right| = \left| \left( \boldsymbol{h}^{[l-1]} \right)^{\mathsf{T}} \boldsymbol{\delta}^{[l]} \right| \tag{8}$$

$$\implies \left| \nabla \theta_{a,b}^{[l]} \right| = \left| \sum_{i=1}^{B} \boldsymbol{h}_{i,a}^{[l-1]} \boldsymbol{\delta}_{i,b}^{[l]} \right| \leq \sum_{i=1}^{B} \left| \boldsymbol{h}_{i,a}^{[l-1]} \boldsymbol{\delta}_{i,b}^{[l]} \right| \tag{9}$$

$$= \sum_{i}^{B} \left| \boldsymbol{h}_{i,a}^{[l-1]} \right| \sum_{i}^{B} \left| \boldsymbol{\delta}_{i,b}^{[l]} \right| - \sum_{r,s \neq r}^{B} \left| \boldsymbol{h}_{i,r}^{[l-1]} \boldsymbol{\delta}_{i,s}^{[l]} \right| \leq \sum_{i}^{B} \left| \boldsymbol{h}_{i,a}^{[l-1]} \right| \sum_{i}^{B} \left| \boldsymbol{\delta}_{i,b}^{[l]} \right| \tag{10}$$

$$\implies \left| \nabla \theta^{[l]} \right| \leq \left( \sum_{i}^{B} \left| \boldsymbol{h}_{i,\cdot}^{[l-1]} \right| \right)^{\mathsf{T}} \left( \sum_{i}^{B} \left| \boldsymbol{\delta}_{i,\cdot}^{[l]} \right| \right) \tag{11}$$

$\square$

## E    SUBSET MULTIPLIER

The extended results on the affect of the size of the subset are shown in Figure 3.

## F    LAYER-WISE SPARSITY

We show the sparsity of each layer obtained with the various methods at the end of training in Figure 4.

## G    IMPLEMENTATION NOTES

The implementation of the SynFlow heuristic required the usage of double floating point precision and normalization of the output criteria in order to prevent the values from overflowing. This is one additional disadvantage of using the SynFlow heuristic, in addition to its requirement to make a copy of the model, making it consume three times the memory of GraBo.

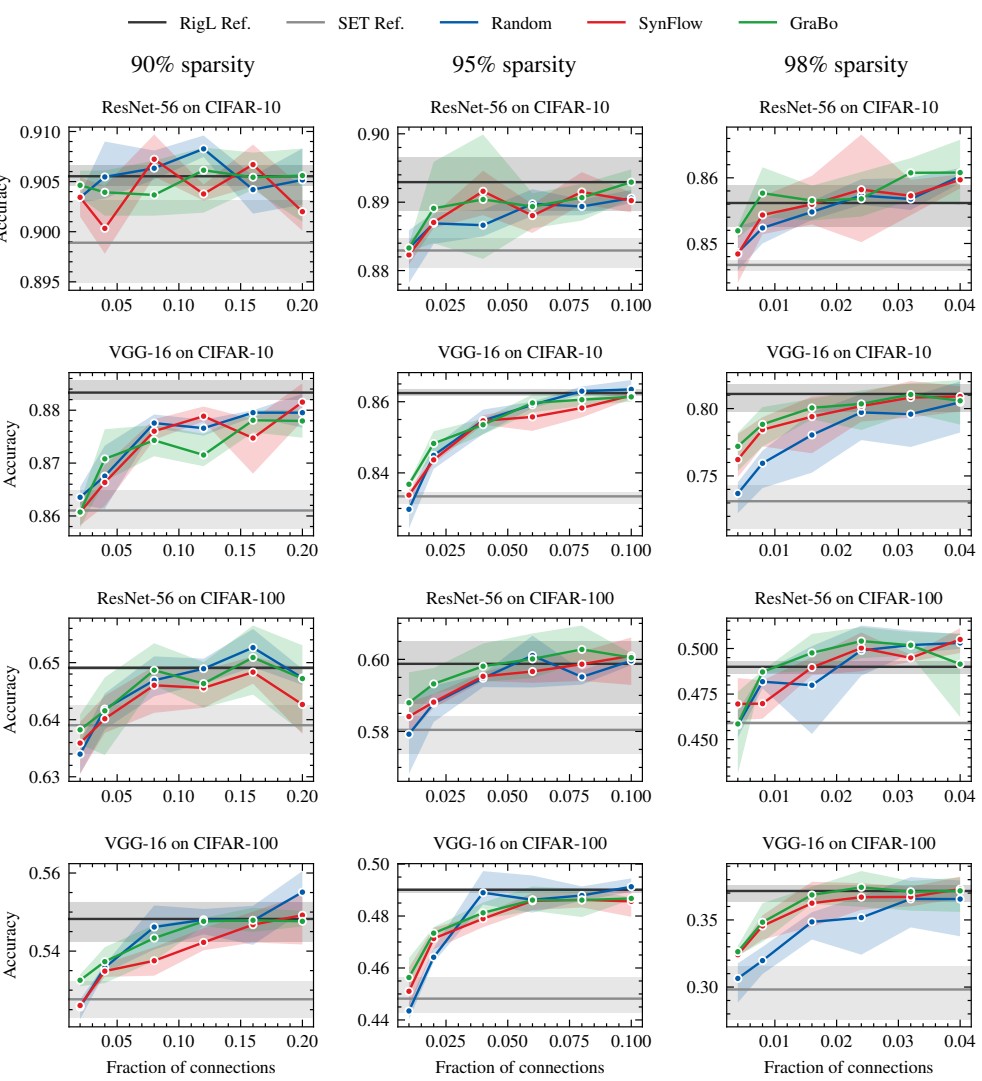

Figure 3: The accuracy of the heuristics while increasing the subset size $\mathbb{S}$ as a fraction of $|\mathbb{W}|$. Showing ResNet-56 and VGG-16 models trained on the CIFAR-10/100 datasets with 90%, 95%, and 98% sparsity.

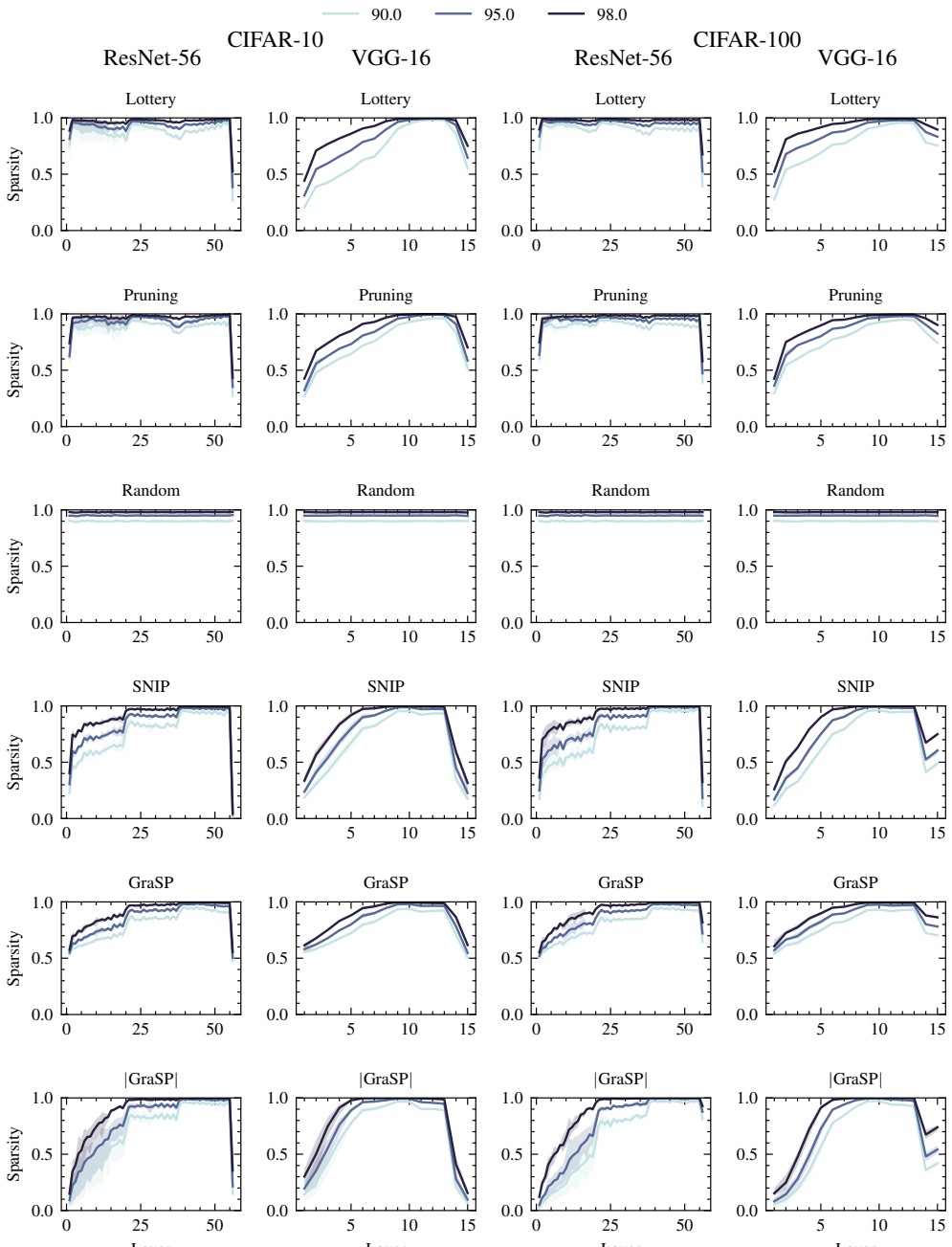

Figure 4: (Part 1) The layer-wise sparsity resulting from each method at the end of training. Showing ResNet-56 and VGG-16 models trained on the CIFAR-10/100 datasets with 90%, 95%, and 98% sparsity.

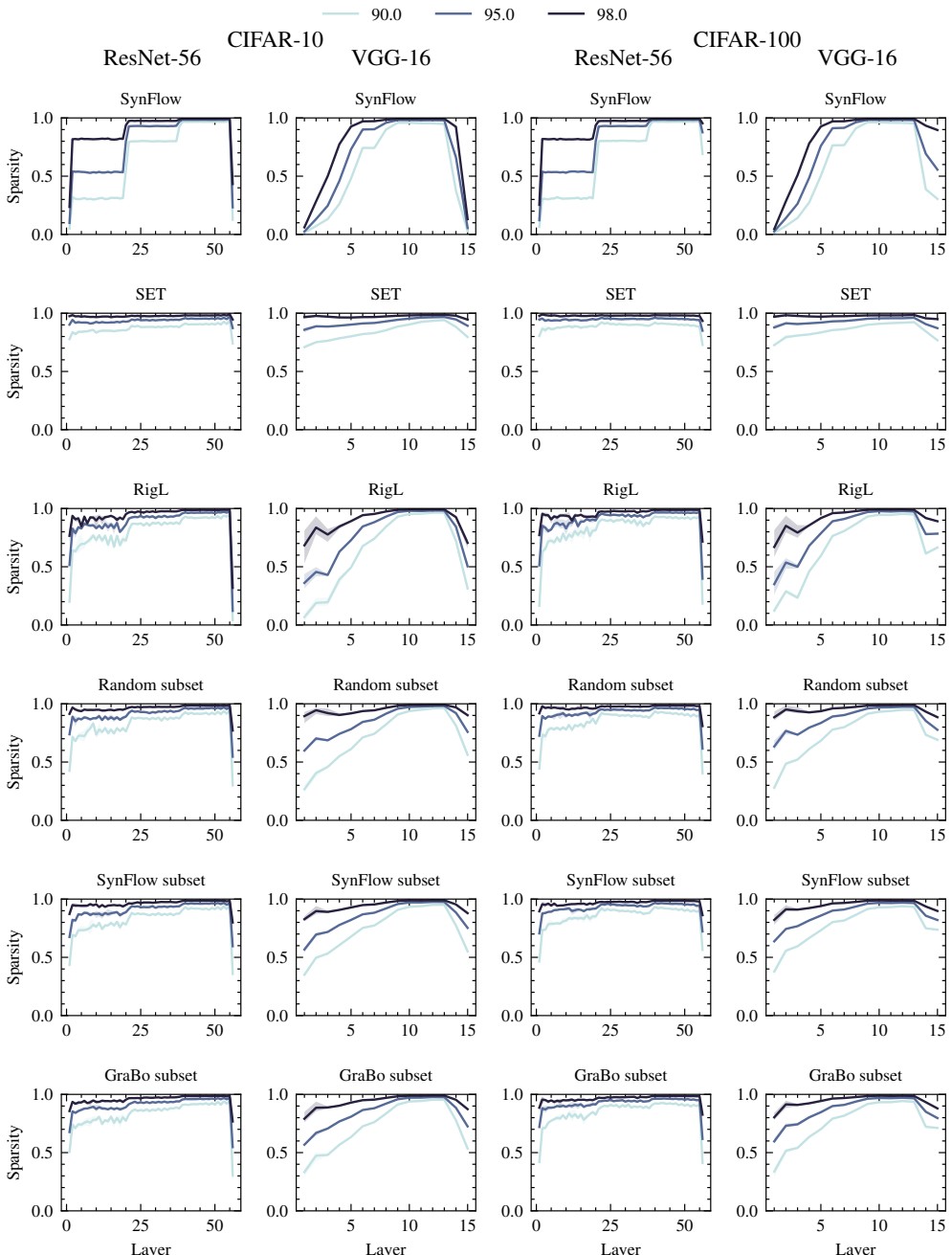

Figure 4: (Part 2) The layer-wise sparsity resulting from each method at the end of training. Showing ResNet-56 and VGG-16 models trained on the CIFAR-10/100 datasets with 90%, 95%, and 98% sparsity. For our method $|\mathbb{S}| = 10|\mathbb{P}|$.

