# OpenReview forum: "Memory Efficient Dynamic Sparse Training"
_ICLR.cc/2023/Conference — Submitted to ICLR 2023_

### Official Review · Reviewer_U17z · 2022-10-22

**Confidence:** 4
**Correctness:** 3
**Technical Novelty And Significance:** 2
**Empirical Novelty And Significance:** 1
**Recommendation:** 3

**Clarity, Quality, Novelty And Reproducibility:**

I think adding a plot or algorithm illustrating the whole process of the proposed method will improve the clarity of this work.

**Strength And Weaknesses:**

Given that large deep learning models are becoming popular, reducing the training memory cost of deep neural networks is an important topic and has high practical value. However, I have several major concerns:
1. Dynamically pruning neural networks during training to reduce the training memory is not new. As far as I know, [1] has explored the same problem.
2. Current experiments are only done on small datasets and neural networks. However, in practice, we only need such memory-saving techniques when using large datasets and large neural networks.
3. It seems that the proposed method prunes connections in a fine-grained manner. If so, there is no saving on common training hardware platforms (e.g., GPU). In the experiments, I did not find any results showing the real memory saving of the proposed method.


[1] Dynamic sparse graph for efficient deep learning, ICLR 2019

**Summary Of The Paper:**

This paper introduces a method for dynamically pruning connections during training based on a heuristic of the gradient magnitude. The goal is to reduce peak training memory consumption while maintaining accuracy. The method is evaluated on CIFAR10 & CIFAR100 using ResNet-56 & VGG-16.

**Summary Of The Review:**

I think reducing training memory is important, but I have several major concerns regarding the novelty, experiments, and practical value of the proposed method (see Strength And Weaknesses).

---

### Official Review · Reviewer_WdYk · 2022-10-24

**Confidence:** 4
**Correctness:** 2
**Technical Novelty And Significance:** 2
**Empirical Novelty And Significance:** 2
**Recommendation:** 3

**Clarity, Quality, Novelty And Reproducibility:**

- Selected models are too big for a given dataset (CIFAR-10 or 100). ResNet-56 is too huge and this reviewer doubts whether such a large model size is essential for the experiments. This reviewer is not sure whether DST works only for those extremely large models. VGG is also too big to be investigated for such tiny dataset. The authors need to include at least some parameter-efficient models (such as ResNet-18) on ImageNet dataset, while experiments on Transformers would strengthen the claim significantly.
- In Eg. (7), computing h and \delta to obtain gradients of sampled connections may lead to overhead in the overall computations. Does training time increase due to Eg. (7) even though the overall goal is to reduce memory footprint.
- Overall, novelty is not impressive, and the overall contribution (explained in Figure 1) seems to be incremental while the claims are not supported by reasonable models.

**Strength And Weaknesses:**

*Strength
- The authors empirically show that sampled connections can achieve a similar model accuracy compared to the case of DST algorithms using all connections.
- Various methods to obtain sampled connections, such as Random, SynFlow, and GraBo, are presented and compared (as shown in Figure 2)
- For a given subset size (i.e., the number of unpruned connections), the authors show that scaling up widths of a model can improve model accuracy.

*Weakness
- Even though a major contribution of this manuscript is from memory saving during training, discussions and practical analysis of memory saving are missing.
- The authors discuss energy consumption saving as a gain to apply the proposed DST algorithm. However, convergence and the overall training time are not analyzed in the manuscript. A thorough and detailed analysis on the overall training results (including training and valid loss graphs) need to be presented.

**Summary Of The Paper:**

The authors propose a new DST algorithm that utilizes only sampled connections to reduce peak memory usage while previous DST algorithms rely on computing gradients of all connections. For some selected models, the proposed method can obtain similar model accuracy using only 4% as a size of sampled connections compared to the case of DST cases that investigate all connections.

**Summary Of The Review:**

- Practical discussions on reduced memory footprint, energy consumption, and the overall training time are missing.
- Model selections are not practical.

---

### Official Review · Reviewer_qDT4 · 2022-10-24

**Confidence:** 4
**Correctness:** 2
**Technical Novelty And Significance:** 3
**Empirical Novelty And Significance:** 1
**Recommendation:** 3

**Clarity, Quality, Novelty And Reproducibility:**

The paper is well written overall. The authors could improve the readability further as follow:
* Explicitly claim what is novel in their approach. The paper does a great job of summing up the state of the art and tying it into their approach. However, as a result, unless someone is already familiar with existing literature, it becomes difficult to spot exactly where previous work ends and new contributions start.
* Better explain the relationship between SynFlow and GraBo.
* Explicitly state the strategy used to prune connections at each step.


**Strength And Weaknesses:**

The stated benefit of this paper is to reduce peak memory usage during training a model. However, there is no measurement or estimation of the memory saved by this new approach saves. The authors blame the lack of comprehensive support for sparse operations in PyTorch, but the authors could have simply sparsified/densified tensors on the fly as needed to work around the limitations of PyTorch (though at the cost of some overhead due to the temporary storage of a dense version of each tensor). This is a major limitation that prevents a proper assessment of the value of the work.

The evaluation is performed on only two CNNs: VGG-16, and ResNet-56. Since the approach is a heuristics, its evaluation needs to be performed on a broader set of DNNs to ensure that it's broadly applicable. For example, I would have liked to see different neural network architectures, such as vision transformers, and different application domains, such as speech, natural language, generative models, etc... PyTorch is shipped with a comprehensive list of models that can be quickly leveraged to strengthen the evaluation.





**Summary Of The Paper:**

Dynamic sparse training (DST) has been proposed to sparsify models during training. However, all the previously published method still required similar amounts of memory to train the sparsified model and the dense model. This paper describe a new technique that overcomes this limitation. Specifically, they contribute:
* a new sampling strategy used to identify a set of pruned connections that should be activated in the next training iteration.
* an empirical evaluation of the approach that compares its impact on model accuracy against a comprehensive set of alternative approaches.

**Summary Of The Review:**

The authors tackle a problem of great importance, namely decreasing the peak memory usage of deep learning. However, I feel that the evaluation of their approach needs to be improved to support the authors claims of reducing memory usage and meeting the performance of existing DST approaches.

---

### Official Review · Reviewer_ghDx · 2022-10-28

**Confidence:** 5
**Correctness:** 2
**Technical Novelty And Significance:** 1
**Empirical Novelty And Significance:** 1
**Recommendation:** 3

**Clarity, Quality, Novelty And Reproducibility:**

See the strengths and weaknesses.


**Strength And Weaknesses:**

Strength:

1. This paper is well written and easy to read.

Weaknesses:

1. The idea of the proposed method is unreasonable. For convolutional neural networks, most of the memory cost in training is spent in storing the activations, which would be scaled up by the mini-batch size, insteads of the weights. In the proposed method, the activations are unlikely to be pruned, therefore the memory cost cannot be saved effectively.

2. The proposed method hasn’t been evaluated properly. The authors need to show the memory cost during the whole training process instead of the size of the sparse model.

3. The authors also need to conduct experiments with large datasets, such as ImageNet, which is a standard dataset for sparse training/pruning studies.


**Summary Of The Paper:**

In this paper, the authors propose a new weight level sparse training method for deep neural networks. Its motivation is to reduce the peak memory cost during training. The idea is to sample a subset of inactive in each iteration and optimize the loss over these weights. The authors conduct some experiments to evaluate the performance of their method. However,  the proposed method also hasn’t been evaluated properly.  More importantly, the idea of this paper is unreasonable as for convolutional neural networks, most of the memory during training is spent by the activations instead of the weights.

**Summary Of The Review:**

1. The idea of the proposed method is unreasonable.

2. The proposed method hasn’t been evaluated properly.

---

### Decision · Program_Chairs · 2023-01-20

**Decision:**

Reject

**Justification For Why Not Higher Score:**

Clear issues with evaluation (only two very old models evaluated), results (no results showing the actual memory savings) and related work (clear related work from 2019 is missed).

**Justification For Why Not Lower Score:**

N/A

**Metareview: Summary, Strengths And Weaknesses:**

While the reviewers appreciated the clarity of the writing and some of the presented results, they were concerned with (a) the overall evaluation, (b) discussion and analysis of actual memory saving. There was no author response. For these reasons I vote to reject. The reviewers have given extremely detailed feedback and I recommend the authors follow / respond to their comments closely before submitting to a future ML venue. If the authors are able to fix these things it will make a much stronger submission.